# Exploring the Occupational Balance of Young Adults during Social Distancing Measures in the COVID-19 Pandemic

**DOI:** 10.3390/ijerph18115809

**Published:** 2021-05-28

**Authors:** Paula Rodríguez-Fernández, Josefa González-Santos, Mirian Santamaría-Peláez, Raúl Soto-Cámara, Jerónimo J. González-Bernal

**Affiliations:** Department of Health Sciences, University of Burgos, 09001 Burgos, Spain; prf0011@alu.ubu.es (P.R.-F.); rscamara@ubu.es (R.S.-C.); jejavier@ubu.es (J.J.G.-B.)

**Keywords:** occupational balance, occupational health, young adults, social isolation, home confinement, pandemic, COVID-19

## Abstract

(1) Background: A balanced life is related to good health in young people, one of the groups most affected by confinement and social distancing measures during the COVID-19 pandemic. This study aims to explore the occupational balance of young adults during home confinement and its association with different sociodemographic factors. (2) Methods: A cross-sectional study was designed, and an online survey was disseminated to collect sociodemographic and occupational balance data, using the Occupational Balance Questionnaire (OBQ). The statistical analysis was performed using the SPSS statistical software package version 24.0. (3) Results: 965 young adults between 18 and 30 years old participated in the study. A predictive model showed that the main predictors of a lower occupational balance were a negative self-perception (β= 0.377; *p* = <0.0001), student status (β = 0.521; *p* = 0.001), not receiving enough information (β = 0.951; *p* = 0.001) and long periods of quarantine (β = 0.036; *p* = 0.007). (4) Conclusions: Considering people’s occupational health and related factors could lessen many of the psychosocial consequences of isolation and contribute to the well-being of young people.

## 1. Introduction

On 11 March 2020, the disease caused by SARS-CoV-2 (COVID-19) was declared a global pandemic [1]. Due to the rapid spread of the virus, which emerged in the Chinese city of Wuhan, public health measures began to be implemented around the world [1]. Physical distancing, social isolation, changes in daily routines, and health risks stemming from the COVID-19 pandemic have triggered serious consequences for the mood and behavior of the general population [1,2,3,4]. The measures implemented to prevent the spread of the virus have proven to be particularly harmful for the younger population [5], especially in young adults and adolescents who seem to be particularly vulnerable to the consequences of the pandemic on mental health [6].

Although the prevalence of mood disorders has increased among young people in recent years [7], it has risen during the last year, leading to a significant decrease in the mood, psychological well-being and quality of life of young adults [8,9,10,11,12,13]. Younger age has been connected to feelings of loneliness during the COVID-19 pandemic, probably due to the disruption of educational, economic, and social life during this public health crisis [14,15]; moreover, the most significant increases in anxiety and depression disorders have been presented in young adult groups [16,17,18]. Depression is characterized by a low mood, lack of energy and decreased activity level, while anxiety manifests itself in the form of restlessness and tension when people suffering from it are exposed to certain everyday situations [19]. Both disorders have serious consequences on people’s lives and lead to difficulties in coping with daily occupations [19].

In this sense, being involved and engaging in a wide variety of occupations is essential to achieve good health and quality of life [20]. Occupational balance is a subjective, multidimensional and health-related phenomenon [21], and depends on the ability of each person to control the distribution of their mental, physical, social, emotional and spiritual capacities and their resources of time, energy, money and materials [22]. It is a dynamic process of balance–imbalance in which new events and circumstances lead to states of imbalance, and individual adaptive capacities allow the recovery or assumption of new occupational roles and patterns to achieve a state of equilibrium. Therefore, the imbalance occurs when people’s health needs such as physical exercise, social relationships and rest, or the needs to do, be, and become, are not covered through participation in daily occupations [23]. Factors such as physical, cognitive, psychosocial and sensorimotor impairments [24], occupational deprivation and occupational injustice [22] were demonstrated to hinder the performance of daily and meaningful occupations in the general population, as well as by suffering from psychological and emotional disorders [25,26,27]. More specifically, one study found low levels of occupational balance in Swedish adults with anxiety and depression [28], and Wagman et al. [23] demonstrated in their research that young adults associate having a balanced life with good health.

Exceptional situations such as the COVID-19 pandemic can cause a far-reaching disturbance in the occupational balance of the general population [29], and despite the fact that a certain imbalance is normal as long as it is not intense or lasts over time, the maintained imbalance is associated with pathological conditions [28]. The relationship between occupational balance and health is supported by some studies [22,25,26,30,31,32], and is associated with quality of life [25], life satisfaction [22,26], decreased perceived stress [31], and good subjective health [32]; it also affirms that a state of occupational balance is the reflection of the harmony between the different dimensions of a human being.

Existing research related to the consequences of the COVID-19 pandemic in the general population has focused mainly on factors associated with mental health, well-being and life satisfaction, but problems in the performance of everyday occupations could have a strong relationship with the aforementioned aspects. Therefore, studying occupational balance of young adults can provide relevant information about this group, which was most affected worldwide impacted globally by the restrictive measures to stop the spread of the virus. Furthermore, occupational balance is proven to be one of the main determinants of health in young people [23], who have been the most affected by measures of social distancing and isolation. We hypothesized that different sociodemographic variables were associated with the occupational balance of young Spanish adults during the forced home confinement. Therefore, the present study aimed to explore the occupational balance of young adults during their forced home confinement and its association with different sociodemographic factors.

## 2. Materials and Methods

### 2.1. Study Design—Participants

A cross-sectional study was designed, that included young adults of both sexes residing in Spain during the months of forced home confinement, with the necessary resources to access the online questionnaire for the research. Participants had to be between 18 and 30 years old, a stage called “young adulthood” according to Erikson’s classification [33], during which time the adult must seek to establish relationships with others to avoid feelings of isolation and separation; very characteristic at this time in life. The sample size and hypotheses were not established prior to obtaining the data.

### 2.2. Procedure—Data Collection

Through a non-probability convenience sample, the study participants were selected and invited to participate in the investigation anonymously and voluntarily. In order to reach the maximum number of subjects during the home confinement stage in Spain, an online survey was designed through Google Forms to obtain the data, and was disseminated through Facebook, Instagram, Twitter and WhatsApp.

Before proceeding with the data collection, participants were informed about the purpose of the study and its voluntary and anonymous nature, requesting signed informed consent in order to continue. The time required to answer the survey was approximately 15 to 20 min and it could be answered from March 16 to May 10, 2020, which was the time period of forced home confinement in Spain. Participants were informed of the possibility to withdraw from the study at any time during the survey, without having to provide any justification. Only fully answered questionnaires were considered for subsequent analysis.

The Bioethics Committee of the University of Burgos approved the research, (Reference IR 14/2020, April 2020), which was carried out in accordance with the ethical principles of the Declaration of Helsinki.

### 2.3. Main Outcomes—Instruments

The primary outcome was occupational balance, which was assessed through the Spanish version of the Occupational Balance Questionnaire (OBQ-E) [34], designed byWagman and Håkansson in 2014 [35]. It is a self-administered questionnaire exploring the balance between different types of activities, their importance, time invested, and perceived satisfaction. It is made up of 13 items valued using a Likert-type frequency scale that ranges from 0 “totally disagree” to 5 “totally agree”, resulting in a total score between 0 and 65, with higher scores indicating a better occupational balance [35]. The questionnaire has good psychometric properties, with relevance and representativeness in the studied population and with enough internal consistency and test–retest reliability [34].

An ad hoc questionnaire was designed to collect sociodemographic data, which was previously piloted in a sample of 15 people who were not part of the subsequent analysis. The following variables were collected: age, days of home confinement, self-perceived health, gender (female/male), residence area (north/south), environment (rural/urban), educational level (primary/secondary/vocational training/university/post-university), employment status (active/unemployed or housework/student), amount of information received (enough/insufficient), infected by Sars-Cov-2 (yes/no), isolation (yes/no), direct contact with COVID-19 (yes/no).

The terms north or south were defined as the autonomous communities above or below the capital, respectively, with the capital included in the northern zone.

### 2.4. Statistical Analysis

The OBQ-E was not normally distributed, therefore, the descriptive statistics were expressed in terms of median ($\bar{x}$) and interquartile range (IQR). Univariate analysis was performed using the Mann–Whitney U test and the Kruskal–Wallis test, if there were three or more response options, to analyze the association between the OBQ-E and the different sociodemographic variables. Spearman’s correlation was used to check the relationship between OBQ-E and other continuous variables.

In order to identify possible factors associated with a lower occupational balance during forced home confinement, a forward stepwise multivariable regression analysis, adjusted by age and gender, was carried out. All variables with a *p*-value of ≤0.05 in the previous analysis were included in the multivariate analysis.

The statistical software package SPSS version 25.0 (IBM SPSS Inc, Chicago, IL, USA) was used to perform the analysis. For the analysis of statistical significance, a value of *p* < 0.05 was established.

## 3. Results

A total of 965 adults aged between 18 and 30 years ($\bar{x}$ = 24 (21–27)) were part of the study, with more women than men (80.21% versus 19.79%). Of the participants, 39.79% had a job at the time the survey was completed, 8.81% were unemployed and 51.40% were students; regarding their academic level, 66.22% confirmed they attended or have attended university. Most of the sample were not infected with the virus (98.86%), not isolated in a room (94.30%) and had no direct contact with COVID-19 (92.54%) during forced home confinement. Participants completed the survey between day 4 and day 57 ($\bar{x}$ = 33 (10–37)) of forced home confinement in Spain. For all participants, the median OBQ-E was 45 (IQR 34–57).

Table 1 and Table 2 show the frequencies and univariate analysis of the OBQ-E based on different sociodemographic variables. Regarding the categorical variables (Table 1), residing in northern Spain (*p* = 0.037) and not receiving enough information about the COVID-19 pandemic (*p* < 0.0001) were individually associated with a worse occupational balance during the home confinement. In addition, people with university studies showed worse results compared to those with post-university studies (*p* = 0.010); regarding their employment status, being a student was associated with lower satisfaction with the number and variation of occupations and with time spent in them compared to active people (*p* < 0.0001).

A statistically significant positive relationship was found between the OBQ-E and continuous variables such as age and self-perceived health; older young adults with a better self-perception of health correlated to better occupational balance (*p* < 0.0001). In addition, the days of home confinement were significantly negatively correlated with the main variable of the study, producing a greater alteration in the occupational balance with the course of forced home confinement (*p* < 0.0001) (Table 2).

Table 3 shows the multivariate analysis carried out with the significant variables from the previous analysis, which produced R² = 0.191, F (_4,960_) = 56,802, *p* < 0.0001. A lower self-perceived state of health (*p* < 0.0001), being a student (*p* = 0.001), not receiving enough information from the media and authorities about the pandemic (*p* = 0.001), and prolonged periods of confinement were revealed as predictors of a worse occupational balance (*p* = 0.007). Residence area, educational level, and age were not significantly associated with the study variable in the multivaried analysis (*p* > 0.05).

## 4. Discussion

The COVID-19 crisis has produced serious mood and behavior changes, probably due to restrictive measures such as social distancing and isolation, changes in aspects of daily life, and the health threat perceived by the population [1,2]. This study is the first to explore the occupational balance of young adults during the pandemic, but despite the lack of research in this field, factors related to mental health [36], lifestyle [13,37], emotional distress [11], loneliness [38], behavior and well-being patterns [12] have been studied in this population. Taking into account the scores obtained in studies on general populations, occupational balance disturbances in young adults during forced home confinement cannot be confirmed, as the median of the present sample is similar to that found in other studies [39]. Despite this, statistically significant associations have been found between the primary outcome variable and some of the independent variables analyzed.

Occupational balance and self-perception of health have been positively correlated in previous research [26,27]. Poor self-perceived health has been shown to be the main predictor of low levels of occupational balance in young Spanish adults during the stage of forced home confinement. In line with our results, Meseguer de Pedro et al. [40] found a significant reduction in self-perceived health, caused by the home confinement situation during the COVID-19 pandemic, which predicted emotional exhaustion in Spanish adults. A low self-perceived health condition was also associated with high anxiety and depression levels in the Chinese population affected by quarantine during the pandemic [41], and promoted psychological distress in adults [42]. Uncertainty was one of the main threats to self-perceived health in previous pandemic crises, and the negative correlation between age and intolerance of uncertainty makes younger people suffer more from the psychosocial consequences of lack of information in general [43]. In the case of COVID-19, the uncertainty derived from the impossibility of knowing in advance the duration of the containment measures, the real risk of infection, the symptomatic manifestations of the virus and the consequences at a personal, economic and social level, has shown to promote the emergence of mental health problems and increase fear of the pandemic [44,45]. Receiving insufficient information from the authorities and the media predicts disturbances of occupational balance in young people, probably due to their low tolerance for uncertainty, increasing the level of psychosocial comorbidity and daily life imbalances [46,47,48].

Poorer mental health has also been reported by groups of students during the COVID-19 pandemic [49]. Distance learning has increased uncertainty, stress and anxiety levels, due to the implementation of new teaching and assessment modalities, sometimes unclear and structured by teachers [50]. Magson et al. [18], suggested in their research that the transition to online learning may have been a particularly significant stressor for young people, and Fiorillo and Gorwood [51] linked the psychosomatic distress of college students with the abrupt change in daily routines and uncertainty about the future during the forced home confinement stage. Students must face big questions related to their future and educational prospects, which makes them especially vulnerable to the consequences of uncertainty and misinformation in times of crisis [3]. These findings are in line with the results obtained in this research, where being a student predicts a lower occupational balance during social isolation and forced home confinement.

Although young adults are most affected by physical distancing and social isolation during the COVID-19 pandemic, students, uninformed youth, and young adults with bad self-perceived health demonstrated lower levels of occupational balance during the home confinement. The establishment of specific communication and unconfined policies are essential to face the loss of occupational balance associated with misinformation and intolerance to uncertainty reported by the younger population, as well as to prevent the psychosocial consequences of the pandemic. In addition, a collaboration between the government and universities would be desirable to provide quality social and health care services aimed at aimed at managing moments of crisis for university students [14].

The limitations of this research include the cross-sectional nature of the study and the consequent impossibility of determining causality inferences between the variables analyzed. Data were collected through an online self-report survey, which could lead to bias in the recruitment of participants. This fact, together with the use of convenience sampling, could produce an unrepresentative population sample. Furthermore, it is possible that relevant study variables have been omitted due to the lack of previous specific research, making it difficult to contrast and discuss the results obtained. Considering the strengths, this is the first research to explore the occupational balance of Spanish young adults during the COVID-19 pandemic and possible risk factors in times of crisis. Likewise, it is worth highlighting the data collection on a large sample and the identification of the predictive factors of a significant interruption in young adults’ lives during home confinement and social isolation.

One of the main factors in promoting the wellbeing and health of the population lies in balancing the occupations of daily life and the nature of them, an aspect significantly disturbed in certain groups of young adults during the COVID-19 pandemic. Occupational balance helps individuals to face the day, depending on the productive occupations of each person, and in times of crisis people can have an adaptive or maladaptive response depending on their personal, contextual, social and emotional resources. In addition to influencing people’s occupational health, maladaptive responses can trigger health, social and emotional problems, so it is essential to design occupation-centered approaches to promote the well-being and health of the most affected groups by different crisis situations, such as the COVID-19 pandemic. Future research is recommended to help understand and study the factors associated with occupational health and its influence on people’s quality of life.

## 5. Conclusions

Occupational balance is one of the principal health determinants in young adults, and despite not having found a significant disturbance in the study sample in general, long periods of home confinement, being a student, not receiving enough information, and having low self-perceived health predict a decrease in occupational balance in this group. When the lives of young people are interrupted or altered due to external factors such as the socio-health crisis of COVID-19, strategies must be designed and implemented focused on their behaviors and lifestyles, without forgetting the key contexts in their development, to prevent or reduce imbalances in occupational performance and their psychosocial repercussions.

Taking into account people’s occupational health, and more specifically that of young adults, could avoid or reduce many of the psychosocial consequences of home confinement and social isolation and contribute to their well-being and quality of life.

## Figures and Tables

**Table 1 ijerph-18-05809-t001:** Comparison of OBQ-E scores according to categorical variables.

Sociodemographic Variables	OBQ-E		
	*n* (%)	$\bar{x}$ (IQR)	*p*-Value
**Gender**			0.247
Female	774 (80.21%)	45 (34–57)	
Male	191 (19,79%)	46 (33–55)	
**Residence area**			0.037
North	769 (79.69%)	45 (33–56)	
South	196 (20.31%)	48 (37–59)	
**Environment**			0.502
Rural	310 (32.12%)	44 (33–58)	
Urban	655 (67.88%)	46 (35–57)	
**Educational level**			0.024
Primary studies	2 (0.21%)	40 (32–48)	
Secondary studies	52 (5.39%)	46.5 (31.75–57.75)	
Vocational training studies	68 (7.05%)	43.5 (37–56)	
University studies	639 (66.22%)	44 (32–56) ^a^	
Post-university studies	204 (21.14%)	49 (38–59) ^a^	
**Employment status**			<0.0001
Active	384 (39.79%)	49 (38–58) ^a^	
Unemployed—Home chores	85 (8.81%)	46 (37–59.25)	
Student	496 (51.40%)	42 (31–56) ^a^	
**Amount of information received**			<0.0001
Enough	312 (32.33%)	49 (37–60)	
Insufficient	653 (67.67%)	43 (32.5–56)	
**Infected by Sars-Cov-2**			0.570
Yes	11 (1.14%)	40 (36–53)	
No	954 (98.86%)	45 (34–57)	
**Isolation**			0.554
Yes	55 (5.70%)	43 (35–54)	
No	910 (94.30%)	45 (34–57)	
**Direct contact with COVID-19**			0.260
Yes	72 (7.46%)	42 (32.25–55)	
No	893 (92.54%)	45 (34–57)	

n: number of patients; OBQ-E: Spanish version of the Occupational Balance Questionnaire; $\bar{x}$: median; IQR: interquartile range; COVID-19: coronavirus disease. ^a^
*p* < 0.05 in the post hoc analysis.

**Table 2 ijerph-18-05809-t002:** Correlation between OBQ-E scores and continuous variables.

Sociodemographic Variables	OBQ-E		
	$\bar{x}$ (IQR)	r Spearman	*p*-Value
Age	24 (21–27)	0.167	<0.0001
Days of home confinement	33 (10–37)	(−0.149)	<0.0001
Self-perceived health	6 (5–6)	0.387	<0.0001

OBQ-E: Spanish version of the Occupational Balance Questionnaire; $\bar{x}$: median; IQR: interquartile range; COVID-19: coronavirus disease.

**Table 3 ijerph-18-05809-t003:** Forward stepwise multivariable regression analysis between sociodemographic variables and OBQ-E.

Independent Predictive Factors	Standard Error	β	T	*p*-Value
Self-perceived health	0.323	0.377	12.936	<0.0001
Employment status	0.521	(−0.110)	(−3.374)	0.001
Amount of information received	0.951	0.095	3.254	0.001
Days of home confinement	0.036	(−0.087)	(−2.686)	0.007

OBQ-E: Spanish version of the Occupational Balance Questionnaire.

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
