# Peer review of "Exploring the Occupational Balance of Young Adults during Social Distancing Measures in the COVID-19 Pandemic"

_ijerph, 2021, doi:10.3390/ijerph18115809_

Round 1
Reviewer 1 Report
- Line 26: Initially, I suggest writing two or three sentences that can contextualize the pandemic caused by COVID-19.
- Line 62: “The relationship between occupational balance and health is supported by some studies [30] ...” . I suggest the inclusion of more studies, and not just the reference 30.
Suggestions:
Håkansson, Carita et al. “Occupational balance, work and life satisfaction in working cohabiting parents in Sweden.” Scandinavian journal of public health vol. 47,3 (2019): 366-374. doi:10.1177/1403494819828870
Wagman, Petra et al. “Occupational balance and its association with life satisfaction in men and women with rheumatoid arthritis.” Musculoskeletal care vol. 18,2 (2020): 187-194. doi:10.1002/msc.1454
Park, Sangmi et al. “Effects of occupational balance on subjective health, quality of life, and health-related variables in community-dwelling older adults: A structural equation modeling approach.” PloS one vol. 16,2 e0246887. 11 Feb. 2021, doi:10.1371/journal.pone.0246887
- Line 76: Present the hypotheses raised by the authors of this study.
- Line 79: Include "both sexes".
- Line 135: Enter the p-value considered statistically.
- Line 124-135: It would be important to present a sample calculation that corresponds to the quantitative of the sample of this study. Including, a reference on which it was based for this calculation.
- Table 2 and 3: In the legend, inform the p value considered statistically.
- Line 225-228: “… cross-sectional nature of the study and the consequent impossibility of determining causality inferences between the variables analyzed. Data was collected through an online self-report survey, which could lead to bias in the recruitment of participants. This fact, together with the use of a convenience sampling, could induce a not necessarily representative of the population sample.” In my opinion, if you do a sample calculation, it will help that this is no longer a limitation.
- Line 237: This is the work force. I find it interesting to highlight more. (The strength).
- Conclusion: “Taking into account people´s occupational health, and more specifically of young adults, could avoid or reduce many of the psychosocial consequences of home confinement and social isolation and contribute to their well-being and quality of life.” given the originality and importance of this work, I think this phrase can be better applied. I suggest rebuilding it, including health promotion strategies and care for these young people, so that they maintain health with quality and without negative symptoms.
Author Response
Consulte el archivo adjunto.

Reviewer 2 Report
I believe that the results section needs to be further developed. The results presented in the article are scarce and of little relevance.
Reviewer 3 Report
113 which wa should read which was
144 Sapain should read Spain
145 – 160: It is not very clear which findings are from Table 2 and Table 3. Revise descriptions according to the findings of Table 2 and 3 and clearly indicate.
In 150 – 153, explanations relating to the data shown in Table 2 are missing. ? Please add.
156 – 160 should be the findings from Table 3. If so, clearly indicate and add a relevant p value in each statement like the previous para.
80.21 % of the study subject out of 965 adults were women. The present study has any suggestion that women suffered more than men under the pandemic and confinement? Should there be a possibility, please state in Discussion.
